# Cluster Content Caching: A Deep Reinforcement Learning Approach to Improve Energy Efficiency in Cell-Free Massive Multiple-Input Multiple-Output Networks

**DOI:** 10.3390/s23198295

**Published:** 2023-10-07

**Authors:** Fangqing Tan, Yuan Peng, Qiang Liu

**Affiliations:** 1Guangxi Key Laboratory of Wireless Wideband Communication and Signal Processing, Guilin University of Electronic Technology, Guilin 541004, China; pengyuan@mails.guet.edu.cn; 2College of Electronic and Information Engineering, Shandong University of Science and Technology, Qingdao 266590, China

**Keywords:** content caching, cell-free massive MIMO, energy efficiency, deep deterministic policy gradient algorithm

## Abstract

With the explosive growth of micro-video applications, the transmission burden of fronthaul and backhaul links is increasing, and meanwhile, a lot of energy consumption is also generated. For reducing energy consumption and transmission delay burden, we propose a cell-free massive multiple-input multiple-output (CF-mMIMO) system in which the cache on the access point (AP) is used to reduce the load on the link. In this paper, a total energy efficiency (EE) model of a cache-assisted CF-mMIMO system is established. When optimizing EE, forming the co-operation cluster is critical. Therefore, we propose an energy-efficient joint design of content caching, AP clustering, and low-resolution digital-to-analog converter (DAC) in a cache-assisted CF-mMIMO network based on deep reinforcement learning. This scheme can effectively cache content in APs and select the appropriate DAC resolution. Then, taking into account the channel state information and user equipment (UE)’s content request preference, a deep deterministic policy gradient algorithm is used to jointly optimize the cache strategy, AP clustering, and DAC resolution decisions. Simulation results show that the energy efficiency of the proposed scheme is 4% higher than that of other schemes without the resolution optimization and is much higher than that of the only AP clustering without the joint design of content caching and channel quality.

## 1. Introduction

Due to the rapid development of smart devices such as smart phones, smart watches, smart robots, and drones, mobile data traffic on wireless networks has experienced terrible growth. IDC estimates that by 2023, there will be 48.9 billion connected devices worldwide [1]. Such a large number of devices will not only generate exabytes of data but will also require massive amounts of content, which will create unprecedented challenges in the upcoming communications field. The capacity of a backhaul link has become the bottleneck of a data-intensive network, and it is necessary to find an efficient way to reduce the backhaul link load to meet the rapidly growing demand for mobile communication.

Caching is a well-known technique used to improve the performance of numerous wired networks, such as content-centric networks [2,3,4]. In cellular networks, caching frequently requesting content at the edge of the network can reduce backhaul costs, reduce access latency and power consumption, and increase throughput. In [5], it is proposed to replace the backhaul link by caching on base stations (BSs). By optimizing the cache strategy, it is possible to serve more users within the limits of download time, which significantly increases throughput. In [6], the caching in the BS can lighten the backhaul traffic load. In order to minimize the overall energy consumption attributed to cache and data transmission including inter-BS and BS-to-server communications, Ref. [7] optimizes the allocation of cache sizes for BSs and service gateways. With the goal of reducing the overall energy consumption of the service, the caching strategy is fine-tuned in [8], where the influence of multicast transmission is considered.

At the same time, to cater to the rising need for traffic data, more and more antennas of base stations and smaller and smaller cell radius will inevitably cause more and more inter-cells interference. To tackle this issue, many research efforts have been made to diminish inter-cell interference [9,10]. Two primary approaches exist: massive multiple-input multiple-output (mMIMO) systems [11] and distributed systems [12]. Within mMIMO systems, BS utilizes the spatial multiplexing yielded by an extensive array of antennas. These antennas are coupled with precoding techniques that can substantially and efficiently mitigate both intra-cell and inter-cell interference among user equipment (UEs). Nonetheless, the uniformity of service quality across all terminals is not guaranteed by the system. The terminal near the BS can enjoy better service due to good channel conditions, while the terminal located at the edge of the cell can only get inadequate quality of service. Within a distributed system, multiple BSs or access points (APs) collaborate by exchanging service data and channel state information (CSI), aiming to minimize inter-cell interference. But distributed systems remain centered around individual cells. Multi-cell collaboration essentially extends the coverage of a single cell. Edge effects continue to impact UE positioned at the periphery of the cell. Therefore, the cell-free massive multiple-input multiple-output (CF-mMIMO) technique has been introduced [13,14], which combines the strengths the aforementioned two systems, namely robust interference cancellation and macro diversity gain. Additionally, it has made some enhancements: (1) It shifts from a cell-centric service model to a UE-centric service model, allowing for potential overlap between distinct AP clusters. (2) There exists a substantial number of APs, wide coverage, and APs are closer to the terminal. Thus, it completely eliminates the concept of a cell.

The essence of the CF-mMIMO system is that the mMIMO system moves the AP closer to the terminals through the integration of fronthaul links and a more frequent utilization of the backhaul links. This will cause a sharp increase in the link load in the CF-mMIMO system, which inevitably results in elevated energy consumption. Therefore, traffic congestion in the fronthaul/backhaul link and high transmission energy consumption constitute the bottlenecks that impede the practical implementation of CF-mMIMO systems. The content caching technology proactively stores data in the cache device and directly transmits the data to the terminal during peak hours without the need to obtain data from the central processing unit (CPU) and core network via the fronthaul/backhaul link. Because the requested content is concentrated in a limited number of popular files [15], the cost associated with its caching is diminishing [16], so content caching proves to be a cost-effective and effective technique in lessening the burden on the link. Building upon this notion, a cache-assisted CF-mMIMO system is introduced in [17]. Moreover, we proposed energy-efficient content of a data caching strategy in CF-mMIMO systems in [18], but only research ideas were provided without experimental validation. Therefore, its total energy efficiency (EE) maximization problem is non-deterministic polynomial-hard (NP-hard) and necessitates solutions through inefficient and non-scalable methods. Additionally, researchers have started to consider the joint optimization of user association and caching strategies [19,20,21]. For example, in [19], the high-density satellite-UAV-terrestrial network scenario is considered, and the initial combination optimization problem is effectively solved using game theory and genetic algorithm for clustering and cache placement, respectively. In [20], for a CF-mMIMO-assisted vehicle edge network, a Deep-Q-Network (DQN) algorithm was proposed to optimize the cache decision for improving the network capacity and content delivery performance. Moreover, two deep reinforcement learning (DRL) methods, single-agent reinforcement learning and multi-agent reinforcement learning, were proposed to solve the joint optimization problem of user association and content cache in CF-mMIMO in [21]. However, most existing research focuses on content caching strategies in edge caching, without considering AP clustering strategies.

On the other hand, a substantial quantity of modules of high-resolution analog-to-digital converters (ADCs) and digital-to-analog converters (DACs) generate a lot of power consumption. To avoid this, it is recommended to use low-resolution ADCs (1–3 bits) in CF-mMIMO networks. This trade-off reduces power consumption while sacrificing spectral efficiency (SE). The work of [22] shows that low-resolution ADCs have better EE than high-resolution ADCs in the uplink of CF-mMIMO systems.

Creating a practical model for the total EE of a cache-assisted CF-mMIMO system, one that is both straightforward to calculate and analyze, while also being amenable to effective optimization, poses a significant challenge. To date, little research has been conducted on cache-assisted CF-mMIMO systems with cache assistance, which encourages the development of this study. The primary contributions of this paper can be outlined as follows:In this paper, a new total EE model of a cache-assisted CF-mMIMO system is established, which has the following advantages: the introduction of low-resolution DAC can improve EE; UE-centric cache deployment can provide a better user experience; and considering the influence of different resolution converters on EE, it is more suitable for practical use;A deep deterministic policy gradient (DDPG) algorithm is proposed to solve the joint optimization problem of content cache, AP clustering, and DAC resolution, and it can find the global optimal decision for maximizing the EE performance in cache-assisted CF-mMIMO networks;We compare and discuss the influence of DAC resolutions, the numbers of UEs, and APs on the EE performance. Moreover, the proposed DDPG method is compared with the benchmark methods, such as clustering based on signal-to-interference noise ratio (SINR) and caching strategies based on content popularity. By exploiting the intelligent design, its EE is not only significantly better than the benchmark (BM) methods but also better than the DDPG method based on the joint content cache and AP clustering.

The rest of the paper is organized as follows. In Section 2, we give a model for the cache-assisted CF-mMIMO system. In Section 3, we propose the total EE model of the cache-assisted CF-mMIMO system and formulate the optimization problem. In Section 4, we present an approach based on DRL. The simulation results and discussion are provided in Section 5. Conclusively, we summarize this paper in Section 6.

## 2. System Model

In this section, the signal model, cache model, and DAC resolution model of the cache-assisted CF-mMIMO network are introduced. For the signal model, it describes the transmitted signal in the cache-assisted CF-mMIMO network’s downlink channel. For the cache model, a content cache mechanism is outlined to enhance the network’s EE. For the low-resolution DAC model, the power consumption generated with different resolutions and the effect on the signal transmission are explained.

### 2.1. Signal Model

Figure 1 depicts an example topology of a dynamic collaborative cluster serving UE in a cache-assisted CF-mMIMO network. We consider a downlink CF-mMIMO, i.e., a cache-assisted CF-mMIMO system encompassing of *M* single-antenna AP and *K* single-antenna UE. Every AP is linked to the CPU via a fronthaul link, while the CPU itself connects to the core network via backhaul links. All APs and UEs are distributed randomly across $S_{a}$ area. And we only focus on downlink transmissions in this paper.

In time-division duplex (TDD) mode, all APs provide identical time/frequency resources to each terminal. Let the channel linking the *m*-th AP and the *k*-th UE be
(1)$$g_{mk}={\left(d_{mk}/d_{0}\right)}^{-\alpha }h_{mk}$$
where $d_{mk}$ denotes the distance between the *m*-th AP and the *k*-th UE, $d_{0}=\min_{m,k}d_{mk}$ is the reference distance, α represents the path-loss exponent (α≥2), and $h_{mk}∼CN(0,1)$ denotes small-scale fading.

Let ℜk denote the set of APs serving the *k*-th UE and ℂm represent the set of UE served by the *m*-th AP. We make the assumption that each UE is ensured service from no more than L(L<M) APs (i.e., |ℜk|≤L,∀k). Therefore, the AP set for all services and the UE set for all services can be represented as ℜ=∪k=1Kℜk and ℂ=∪m=1Mℂm, respectively. Let qk be a symbol emitted in the service AP for the *k*-th UE, where E[|qk|2]=1,E[qk]=0,∀k and E[qkql∗]=0,∀k≠l (i.e., the symbols of distinct UE are not related). Then, the transmitted signal of the *m*-th AP can be expressed as [14]
(2)xm=∑k∈ℂmpmkg^mk*qk
where pmk signifies the power assigned to the *k*-th UE at the *m*-th AP subject to power constraints, E[|xm|2]≤Pm is constrained by the maximum power Pm transmitted by the *m*-th AP, and
g^mk
denotes the channel estimation gmk at the *m*-th AP. This paper considers perfect CSI (i.e., g^mk=gmk,∀m,k).

Accordingly, the *k*-th UE’s received signal can be expressed as [23]
(3)rk=∑m∈ℜgmkxm+wk =∑m∈ℜkgmkxm+∑m∈ℜkcgmkxm+wk =∑m∈ℜk∑k′∈ℂmpmk′gmkg^mk′*qk′+∑m∈ℜkcgmkxm+wk =∑m∈ℜkpmkgmkg^mk*qk︸useful signal+∑m∈ℜk′∑k′∈ℂ,k′≠kpmk′gmkg^mk′*qk′+wk︸interference plus noise
where wk∼CN(0,σw2) signifies the noise at the *k*-th UE, and ℜkc=ℜ\ℜk is the other AP set that does not serve the *k*-th UE.

### 2.2. Caching Model

We consider a limited file library $CF=\{cf_{1},cf_{2},\cdots ,cf_{F}\}$ with *F* content files. Let $CF_{m}\subset CF$ be the set of content files cached in the *m*-th AP. Additionally, we make the assumption that each AP has the capacity to cache a maximum of N(N<F) files, denoted as $\left|CF_{m}\right|\leq N,\forall m$. Each UE makes content file requests independently or abandons the request. The content file for the *k*-th UE request is represented by $cf_{k}\in CF$, where $cf_{k}$ is determined by the content preference vector for the *k*-th UE (arranged in descending order of preference for all content files) and the distribution of content popularity (specified with the Zipf distribution). To be more specific, in the content preference vector of the *k*-th UE, the probability that $cf_{k}$ equals the content file of the *i*-th rank is $i^{-\beta }/\sum_{j=1}^{F}j^{-\beta }$, where β is the Zipf factor; Usually, set to β=0.5,1,2. Each UE possesses a different, independent, and time-invariant content preference vector.

We use Hmk to define the event that the content file requested by the *k*-th UE is cached on its *m*-th service AP, i.e., cfk∈CFm,m∈ℜk. Therefore, the matching event of the *k*-th UE Hk indicates that the file requested by the *k*-th UE is cached across all APs serving the *k*-th UE, i.e., cfk∈CFm,∀m∈ℜk. In case of a miss, there exist certain m∈ℜk APs that do not cache the file of the *k*-th UE, i.e., cfk∉CFm,∃m∈ℜk. In such scenarios, these APs will necessitate requesting the content file cfk from the CPU/core network for joint AP transfer. The network’s hit ratio is denoted as H=∑k∈ℂ1Hk/|ℂ|, where 1Hk signifies the indicator function, and it is set to be 1 if the Hk event occurs; otherwise, it is set to 0.

### 2.3. Low-Resolution DAC Model

We adopt a low-resolution DAC with a binary-weighted current-oriented topology, whose power consumption is composed of both static and dynamic components. The power consumption of a DAC module with a resolution of *b* can be given as [24,25]
(4)$$P_{DAC}(b,F_{s})=1.5\times 10^{-5}\cdot 2^{b}+4.5\times 10^{-12}\cdot b\cdot F_{s}$$
where $F_{s}$ is the sampling frequency.

Each AP’s antenna is connected to a low-resolution DAC, and the resulting signal has α∈[0,1] linear gain. Therefore, the transmitting signal of the *m*-th AP given in (2) is now modified as
(5)xm=αm∑k∈ℂmpmkg^mk*qk
where αm represents the linear gain of the *m*-th AP, and its expression is [22,26]
(6)αm={0.6366   ,bm=10.8825   ,bm=21−(π3/2)⋅2−2bm ,bm≥3

The received signal of *k*-th terminal provided by (3) is rewritten as
(7)rk=∑m∈ℜkαmpmkgmkg^mk*qk︸useful signal+∑m∈ℜk′αm∑k′∈ℂ,k′≠kpmk′gmkg^mk′*qk′+wk︸interference plus noise

## 3. The EE Model and Problem Formulation

### 3.1. The System Sum Rate

According to the Shannon theory, the achievable rate of the *k*-th UE can be expressed as
(8)Rk=Blog2(1+|∑m∈ℜkαmpmkgmkg^mk*|2∑k′∈ℂ,k′≠k|∑m∈ℜαmpmk′gmkg^mk′*|2+|wk|2)
where *B* is the bandwidth. Therefore, the overall achievable rate of the considered cache-assisted CF-mMIMO network is given using
(9)Rsum=∑k∈ℂRk

### 3.2. Power Consumption

The overall power consumption of the network consists of four parts: (1) the transmission power of all service APs; (2) power consumption of DAC in all service APs; (3) the power required by the AP to recover the lost content file from the CPU; and (4) the power needed by the CPU to recover the lost content from the core network.

For (1), the total transmitted power of all service APs is represented by $\sum_{m\in ℜ}P_{m}$. For (2), the sum of DAC power consumption in all service APs is given using $\sum_{m\in ℜ}P_{DAC}^{m}$, where
$P_{DAC}^{m}$
indicates the power consumption of the DAC module with $b_{m}$ resolution selected using the *m*-th AP.

For (3) and (4) in cache-assisted CF-mMIMO systems, all APs within a cluster must concurrently transmit identical content to the terminal. Cache deployment results in three scenarios are illustrated in Figure 2: (a) Every AP within the cluster has deployed the required content. (b) Only some APs have deployed the required data. c) None of the APs deployed the required content. In Scenario (a), no content is conveyed through either the fronthaul or backhaul link. In Scenario (b), certain APs are required to transmit content via the fronthaul link. In Scenario (c), all content is transmitted to the AP via the backhaul link of the core network and the fronthaul link of the CPU.

The fronthaul link is utilized for content transmission between the AP and the CPU, its power consumption is proportional to the cumulative SE sum, and its expression is [27]
(10)Pbh,m=Ebh∑k∈ℂmRk
where Ebh indicates the energy consumed for transmitting 1 Mbit of data over the fronthaul link.

The *m*-th AP is used to transmit data $q_{1},q_{2},\cdots q_{K}$ via a fronthaul/backhaul link between the CPU and core network. Therefore, the fronthaul/backhaul power consumption depends on the SE, ${\mathrm{SE}}_{1},{\mathrm{SE}}_{2},\cdots {\mathrm{SE}}_{K}$. If the *m*-th AP serves only specific UEs, it merely transmits data related to these UEs. Therefore, the power consumption for the fronthaul/backhaul is contingent solely on the SE of these UEs. As shown in Figure 2, the cache power consumption is calculated with the user as the center, so the fronthaul power consumption of the *k*-th cluster can be represented as
(11)$$P_{bh,k}=E_{bh}R_{k}$$

Similarly, the backhaul power generated by the *k*-th cluster’s backhaul link for transferring data between the core network and the CPU can be expressed as
(12)$$P_{bb,k}=E_{bb}R_{k}$$
where $E_{bb}$ indicates the energy consumed for transmitting 1 Mbit of data over a backlink. Therefore, the power uploaded via the AP to the CPU in the *k*-th cluster can be represented as
(13)$$P_{bh,k}^{up}=\left\{ \begin{array}{l} P_{bh,k},\;0<\sum_{m\in {ℜ}_{k}}\left|H_{mk}^{miss}\right|/\left|{ℜ}_{k}\right|<1\\ \;\;0,\;\mathrm{others} \end{array}\right.$$
where $H_{mk}^{miss}=\left|1-H_{mk}\right|$ represents the event that the UE content request provided by the *m*-th AP is not cached on the *m*-th AP. The value is 1 if the event
$H_{mk}^{miss}$
occurs, but it is 0 otherwise.

The energy consumption associated with the content requested by the AP from the CPU in the *k*-th cluster can be expressed as
(14)$$P_{bh,k}^{down}=P_{bh,k}\sum_{m\in {ℜ}_{k}}\left|H_{mk}^{miss}\right|$$

The backhaul power generated by the *k*-th cluster requesting content from the core network can be denoted as
(15)$$P_{bb,k}^{down}=P_{bb,k}\left\lfloor \sum_{m\in {ℜ}_{k}}\left|H_{mk}^{miss}\right|/\left|{ℜ}_{k}\right|\right\rfloor$$

Therefore, the fronthaul/backhaul power consumption of the *k*-th cluster can be expressed as
(16)$$P_{B,k}=P_{{}_{bh,k}}^{up}+P_{{}_{bh,k}}^{down}+P_{{}_{bb,k}}^{down}$$

So, the overall energy consumption can be expressed as
(17)$$P_{total}=\sum_{m\in ℜ}\left(P_{m}+P_{DAC}^{m}\right)+\sum_{k=1}^{K}P_{B,k}$$

### 3.3. Problem Formulation

Our aim is to find a strategy that determines the AP clustering with ℜ1,ℜ2,⋯,ℜK, as well as the AP’s content cache CF1,CF2,⋯,CFM and its DAC resolution b1,b2,⋯,bM, in order to maximize the system’s EE. The optimization problem can be expressed as
(18)max RsumPtotals.t.  (C1):|ℜk|≤M,∀k∈ℂ,    (C2):|CFm|≤L,∀m∈ℜ,    (C3):bm∈N+,∀m∈ℜ.
where constraint (C1) indicates that the count of APs within the AP cluster of each UE cannot exceed the maximum number of connections *M*. Constraint (C2) requires that the amount of content cached on each AP must not exceed its maximum capacity number *L*. Constraint (C3) means that the resolution of each DAC is a positive integer.

To maximize the EE performance, the design for trade-offs is needed. First, AP clusters based on channel quality and high-resolution DAC can select better channels to get the best SE. In contrast, an AP cluster based entirely on cached content and low-resolution DAC can avoid the energy consumption of the fronthaul/backhaul link, reducing the energy consumption of the DAC module. In addition, in large networks, solving this issue is complicated due to the large number of APs and UEs. To solve this problem, we developed deep reinforcement learning-based content caching, AP clustering, and DAC resolution co-selection strategies, which will be elaborated upon in the Section 4.

## 4. Deep Reinforcement Learning Method

In this section, we will describe how the DDPG algorithm solves the combined predicament of AP clustering, caching, and selecting DAC resolution. Three basic components (action, state, and reward) are defined in reinforcement learning (RL) problems.

### 4.1. Action, State, and Reward

In slot *t*, action $a_{t}$ encompasses the processes of clustering, caching, and selecting resolution. Let $a_{mk,t}\in \{0,1\}$, $a_{mcf,t}\in \{0,1\}$, and $a_{mb,t}\in \{0,1\}$ represent the status of *m*-th AP and *k*-th UE service, the *cf* file cache, and the *b* bit resolution switch, respectively, where “1” indicates that the service or cache is successful or enabled and “0” indicates that the service or cache is not served, there is no cache, or it is disabled. So, action $a_{t}$ can be defined as
(19)$$a_{t}≜\left\{a_{t}^{cl},a_{t}^{ca},a_{t}^{res}\right\}$$

The sets $a_{t}^{cl}=\{a_{mk,t}:m\in M,k\in K\}$, $a_{t}^{ca}=\{a_{mcf,t}:m\in M,cf\in CF\}$, and $a_{t}^{res}=\{a_{mb,t}:m\in M,b\in N^{+}\}$ contain aggregate results representing the *t*-th time slot for clustering, caching, and selecting resolution, respectively.

Similarly, the action *a_t_* uniquely determines the sets CFm, bm, ℜk and ℂm, i.e., CFm={cf:amcf,t=1,cf∈CF}, bm={b:amb,t=1,b∈N+}, ℜk={m:amk,t=1,m∈M}, and ℂm={k:amk,t=1,k∈K}.

The state considered in RL should be the set of information that the CPU can collect to compute the reward. In this article, the state of the *t*-th slot is characterized as the collection of channel gain $G_{t}=\{g_{mk,t}:m\in M,k\in K\}$, the action of the preceding time slot, and the historical record of file requests for each UE. Define the history set of user requests as $e_{t}=\{e_{kcf,t}:k\in K,cf\in CF\}$, where $e_{kcf,t}=\sum_{t^{\prime }=1}^{t}1_{cf_{k,t^{\prime }=cf}}$ is the *cf* file download from the *k*-th UE request as of time *t*. So, the state can be denoted as
(20)$$s_{t}≜\left\{G_{t},a_{t-1},e_{t}\right\}$$

According to the objection function of the optimization problem (18), the reward function of the *t*-th slot is defined as
(21)$$r\left(s_{t},a_{t}\right)≜\frac{R_{sum,t}}{P_{total,t}}$$
where $R_{sum,t}$ and $P_{total,t}$ are given in (9) and (17), correspondingly, and the extra subscript *t* is given to emphasize the dynamic behavior. It is worth noting that the total achievable rate $R_{sum,t}$ is contingent upon the channel conditions $G_{t}$, the clustering outcome $a_{t}^{cl}$, and the result $a_{t}^{res}$ of the selection resolution, while the overall power $P_{total,t}$ is contingent upon the caching result $a_{t}^{ca}$ and the result $a_{t}^{res}$ of the selection resolution.

### 4.2. Deep Deterministic Policy Gradient Approach

DDPG algorithm utilizes an actor–critic network architecture. Moreover, each network is accompanied by its respective target network, resulting in a total of four networks within the DDPG algorithm, namely, the actor network $\mu (\cdot |\theta^{\mu })$, critic network $Q(\cdot |\theta^{Q})$, target actor network $\mu^{\prime }(\cdot |\theta^{\mu^{\prime }})$, and target critic network $Q^{\prime }(\cdot |\theta^{Q^{\prime }})$. Each network updates according to its own update rules, maximizing cumulative expected returns. Figure 3 gives the schematic diagram of the DDPG algorithm.

The DDPG algorithm is well-suited for multi-task learning, aligning with the objectives of this paper. This algorithm enhances training stability by adopting a deterministic strategy, which means it directly outputs a specific action value instead of a probability distribution. The algorithm is trained using an experience replay buffer to store past experiences and then to randomly sample from it. This approach breaks the data correlations and ensures that the data conform to an independent distribution, thereby reducing parameter update variance and enhancing convergence speed. Additionally, experiences can be reused, resulting in high data utilization. DDPG leverages neural networks to represent policies (actor) and value functions (critic), making it suitable for high-dimensional state spaces and capable of learning from vast amounts of perceptual data. In comparison to the widely used DQN, DDPG is particularly apt for continuous action spaces. Furthermore, employing actor networks can improve training efficiency, and having more target actor networks and target critic networks helps prevent potential overestimation issues present in DQN.

Algorithm 1 primarily revises the parameters of the actor network and critic network. The actor network adjusts the weight $\theta^{\mu }$ by aiming to maximize the cumulative anticipated reward. The critic network adjusts the weight $\theta^{Q}$ by seeking to minimize the discrepancy between the evaluation value and the target value. Regarding the update process of the target network, a soft update method is adopted, which can also be called exponential average motion. That is, the learning rate (or momentum) τ is introduced, and the weighted average of the previous target network parameters and the current corresponding network parameters are subsequently applied to update the target network. Algorithm 1 summarizes the whole DDPG algorithm process.
**Algorithm 1** DDPG Algorithm Procedure$1:\;\mathrm{Actor}–\mathrm{critic}\;\mathrm{network}\;\mathrm{parameter}\;\theta^{\mu }$$\;\mathrm{and}\;\theta^{Q}$ initialization$2:\;\mathrm{Set}\;\mathrm{the}\;\mathrm{same}\;\mathrm{parameters}\;\theta^{\mu^{\prime }}$$\;\mathrm{and}\;\theta^{Q^{\prime }}$ in the target network3: **for** plot = 1 to *Plot* **do**4:   **for** timeslot = 1 to *T* **do**$5:\;\;\;\;\mathrm{Generate}\;\mathrm{action}\;a_{t}$$\;\mathrm{through}\;\mathrm{the}\;\mathrm{actor}\;\mathrm{network}\;\mu (s_{t}|\theta^{\mu })$$6:\;\;\;\;\mathrm{Get}\;\mathrm{rewards}\;r\left(s_{t},a_{t}\right)$$\;\mathrm{and}\;\mathrm{next}\;\mathrm{status}\;s_{t+1}$$\;\mathrm{according}\;\mathrm{to}\;\mathrm{the}\;\mathrm{action}\;a_{t}$$7:\;\;\;\;\mathrm{Get}\;\mathrm{the}\;\mathrm{target}\;\mathrm{value}\;q\;\mathrm{through}\;\mathrm{the}\;\mathrm{critic}\;\mathrm{network}\;Q(s_{t},a_{t}|\theta^{Q})$$8:\;\;\;\;\mathrm{Use}\;\mathrm{the}\;\mathrm{target}\;\mathrm{network}\;Q^{\prime }(s_{t+1},{a^{\prime }}_{t+1}|\theta^{Q^{\prime }})\;\mathrm{to}\;\mathrm{get}\;\mathrm{the}\;\mathrm{separate}\;\mathrm{target}\;\mathrm{value}y$9:    The gradient is determined by the target value q oftheactor–criticnetworkand the target network target value y10: $\;\;\;\mathrm{Update}\;\mathrm{parameters}\;\theta^{\mu }$$\;\mathrm{and}\;\theta^{Q}$ in the network of actors and critics according to the gradient11: $\;\;\;\mathrm{Update}\;\mathrm{parameters}\;\theta^{\mu^{\prime }}$$\;\mathrm{and}\;\theta^{Q^{\prime }}$ in the target network according to the$\;\mathrm{parameters}\;\theta^{\mu }\;\mathrm{and}\;\theta^{Q}\;\mathrm{in}\;\mathrm{the}\;\mathrm{actor}\;\mathrm{and}\;\mathrm{critic}\;\mathrm{network}\;\mathrm{and}\;\mathrm{the}\;\mathrm{learning}\;\mathrm{rate}\;\tau$
12:   **end for**13:  **end for**

### 4.3. Computational Complexity

In the DDPG algorithm, the input dimension of the neural network is denoted as Input≜M(3K+N+1)+KN, the output dimension is denoted as Output≜M(K+N+b), and the number of model parameters is denoted as Number≜5Input(Input+1)+9Output(Output+1)+10Iuput∗Output, determined by the neural network’s layer count and layer size. The experience pool’s size is Batch=128, and it holds states, actions, rewards, and next states, resulting in a complexity of O(Batch∗(2K(M+N+b)+K(M+N+1)+1)+Number). The decision-making process for actions has a time complexity of O(K+2M), thus leading to the complexity of O(timeslot∗(Number+K+2M)), where timeslot stands for the number of training iterations.

## 5. Simulation Results

### 5.1. Simulation Settings

In this section, we conduct a comparison and analysis of (1) the EE performance of the proposed RL method with three different BM strategies (called BM1, BM2, and BM3), (2) the convergence behavior of DDPG algorithm, (3) the effect of DAC resolutions on the EE, (4) the impact of the number of UE-associated AP on the EE, and (5) the influence of UE and AP quantity on the EE. The BM strategies are given as follows:
BM1: clustering policy based on the SINR (l≤L APs to which the *k*-th UE is connected is the *l* with the highest SINR) and caching policy based on local popularity (in the UE served by the *m*-th AP, the most popular *N* files are cached on the *m*-th AP);BM2: clustering strategy based on SINR (same as BM1) and caching strategy based on network popularity (caching the *N* most popular files across all UEs in all APs);BM3: cache-based clustering strategy (each UE is connected to l≤L APs, and its cache is the content request that best matches each UE in the previous slot) and network-based caching strategy (same as BM2).

The computational complexity of BM strategies is all equal to O(K(M+N+1)), with BM2 having the minimum time complexity of $O((M\log_{2}M)*(KN\log_{2}N))$, BM3 following with $O((KM\log_{2}M)*(KN\log_{2}N))$, and BM1 having the maximum complexity of $O((M\log_{2}M)*(MKN\log_{2}N))$. Although the complexity of the BM strategy is lower than that of DDPG algorithm, the optimization effect of the DDPG algorithm is much better than that of the BM strategy.

We contemplate a situation where APs and UEs are randomly distributed within the region of $S_{a}=1\;km^{2}$, one AP is located at the reference coordinate (0,0). Both the positions of UEs and APs remain constant during the training phase. We set the number of APs and UEs to be M=10 and K=5, correspondingly, the cache size to $\left|CF_{m}\right|\leq 2$ for each AP, the number of files to |CF|=10, and the DAC resolution to $b_{m}\leq 5$. Refer to [14,28,29,30,31,32] for other system settings and parameters, which are summarized in Table 1.

### 5.2. Numerical Results Analysis

Figure 4 shows the convergence of RL+DAC and RL in Algorithm 1, where the EE values versus training episodes are demonstrated. The diagram is trained 1000 times and then fused together. When the BM strategy is compared at the 10th episode of training, it can be seen that the EE of RL+DAC and RL proposed after the 26th episode of training is better than that of other BM strategies. The EE of RL+DAC also completely outperformed the RL algorithm after about the 150th episode of training. Note that in Algorithm 1, RL+DAC assumes that each AP employs a distinct ADC resolution, which sacrifices some computational time in exchange for the improved performance. In contrast, RL employs the same ADC resolution for all APs, thereby reducing the algorithm’s complexity.

Figure 5 illustrates the impact on total EE of changes in the relative positions of APs and UEs over time. It can be easily observed that the total EE of the RL+DAC and RL algorithms are always better than that of other BM strategies. In BM schemes, we find that when each UE is attended to by a single AP at moments 0, 3, 4, 7, 8, 10, and 11, their total EE values are higher than others. When each UE is attended to by three AP at moments 1, 5, 6, and 9, the total EE performance is the highest. What this means is that one UE does not choose more AP services to get a better EE performance. Furthermore, the increasing of the number of UEs will result in a greater number of UEs being served by the APs. Consequently, this necessitates APs to make trade-offs when selecting DAC resolution with the RL+DAC algorithm, significantly diminishing the SE improvement for UEs. When there is an abundance of UEs, this effect becomes nearly equivalent to the average SE achieved in the case of employing the same low resolution at each AP.

Figure 6 depicts the correlation between the number of UEs and the mean SE, where the number of APs is M=10. As illustrated in Figure 6, the mean SE of UEs diminishes as the quantity of UEs increases and then tends to be stable. This phenomenon arises due to the escalation in the quantity of UEs, leading to a gradual intensification of inter-UE interference. Ultimately, the average SE will become stable. Furthermore, the upsurge in the number of UEs will result in a greater number of UEs being served by the APs. Consequently, this necessitates APs to make trade-offs when selecting DAC resolution within the RL+DAC algorithm, significantly diminishing the SE improvement for UEs. When there is an abundance of UEs, this effect becomes nearly equivalent to the average SE achieved when each AP in the RL algorithm utilizes the same low resolution.

Figure 7 explores the influence of the number of UEs on the total EE, where the number of APs is M=10. As observed in Figure 7, the overall trend of the total EE decreases as the quantity of UEs increases and then tends to stabilize. This is because the growth of the number of UEs in the early stage is approximately proportional to the energy consumed by the system, and the existence of inter-UE interference will slow the growth of its sum achievable rate, so its total EE continues to decline. When the number of UEs is large, all APs are already serving UE, and augmenting the number of UEs will not lead to an elevation in the power consumption of AP activities, resulting in a smaller increase in total power consumption, so the total EE will tend to balance. It also indirectly validates the result of Figure 6: the increase in the number of UE does not always guarantee better overall system performance. In other words, the higher the number of UE, the interference between UEs will be particularly significant. Note: the total EE of K=3 UEs in Figure 7 is lower than the total EE of K=4. This is due to the different location of AP and UE, which will lead to different channel conditions, so that the total EE will produce a certain range fluctuation when the quantity of UE is determined. When the quantity is smaller, the fluctuation due to the different effects of the location will be greater.

Figure 8 shows the correlation between the quantity of APs and the sum achievable rate, where the number of UEs is set as K=5. It is readily noticeable that the sum achievable rate firstly increases with the number of APs and then tends to be stable. This is because when the number of APs is small, the UE selects the APs with better channel conditions, so that the rate can be increased. Nevertheless, in scenarios where the number of APs is large, the augmentation for the number of APs brings about a gradual intensification of interference between APs. Consequently, when the number of APs is already relatively high, further increasing the number of APs will not lead to an increase in the sum achievable rate; in fact, it might even decrease it. Moreover, as depicted in Figure 8, the curves for BM1 (l=3) and BM2 (l=3) overlap due to their shared clustering policies, distinct caching strategies, and the fact that the sum achievable rate is solely contingent on SINR and not influenced by caching.

Figure 9 shows the impact of the total quantity of AP on the total EE, where the quantity of UE is K=5. The simulation diagram also indirectly verifies the result of Figure 8, i.e., the more AP, the overall system performance is not necessarily better. At the same time, it illustrated from Figure 9 that when the quantity of APs is 16, the total EE of the system is the highest. Because as the quantity of APs increases, so does their power consumption, and when their sum achievable rate increases slowly, their total EE will begin to decrease. In addition, note that the sum achievable rate and total EE of 20 for the number of AP in Figure 8 and Figure 9 do not strictly follow the trend. This is caused by fluctuations due to the randomness of the positions of APs and UEs.

In Figure 10, the impact of low-resolution DAC on the total EE is demonstrated, and it can be observed that when b≥6, the total EE decreases as the resolution b increases. This means that resolution b will achieve a better total EE performance in the interval [1,5], and the RL+DAC resolution in the figure is $b_{m}\leq 5,\forall m\in ℜ$, which has the best total EE. Therefore, this also validates the wisdom of limiting the resolution range to b≤5 in our RL+DAC algorithm design. In addition, with the increasing resolution b, its total EE decreases faster and faster because in Formula (4), part of the DAC module’s power consumption increases exponentially with the increase in resolution b, while in Formulas (6) and (8), with the resolution b>5, it becomes evident that the sum achievable rate tends to be stable.

## 6. Conclusions

In this paper, an innovative and practical total EE model of a cache-assisted CF-mMIMO system is established. To maximize the total EE, an energy-efficient joint design of content cache, AP clustering, and low-resolution DAC is carried out, and then, a DRL algorithm (i.e., DDPG method) is proposed. Numerical results show that the total EE of the RL+DAC strategy considering DAC resolution is generally 4% higher than that of the RL strategy, and the total EE of these strategies is much higher than those of the BM strategies. In addition, it can be expected that for multi-antenna APs, the number of DAC modules increases linearly with the increase in the quantity of antennas, so the total EE of our proposed RL+DAC strategy will be much higher than that of the RL strategy.

## Figures and Tables

**Figure 1 sensors-23-08295-f001:**
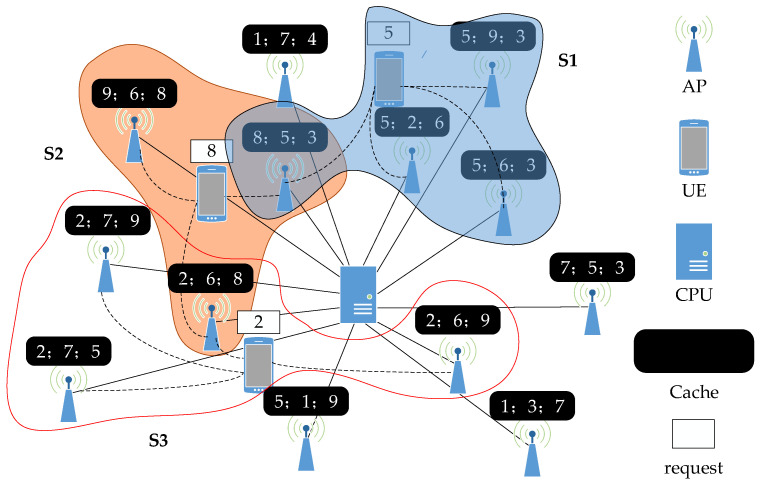
Caching-assisted cell-free massive MIMO model.

**Figure 2 sensors-23-08295-f002:**
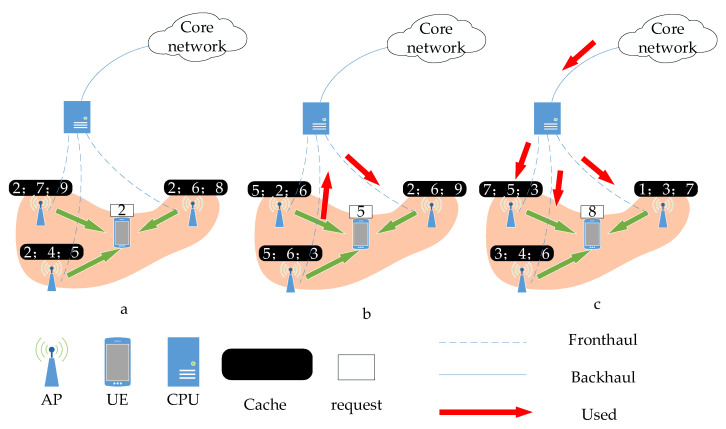
Three scenarios in which content is requested during transmission.Green arrows represent the access link between APs and UEs, red arrows represent the backhaul/fronthaul links between the core network and CPU or APs and CPU.

**Figure 3 sensors-23-08295-f003:**
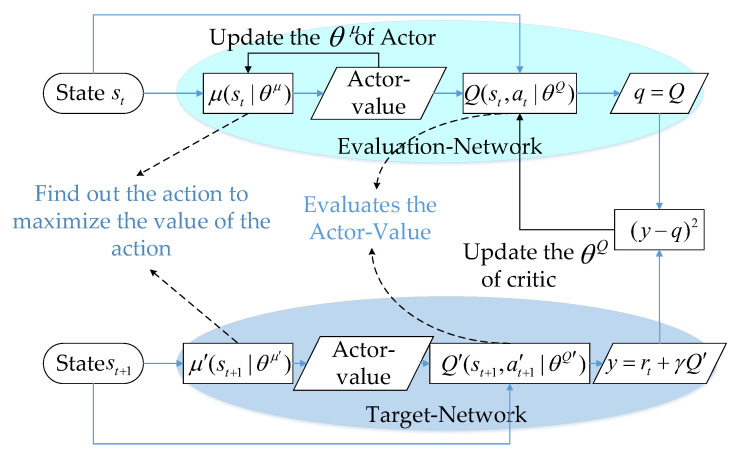
The principle of DDPG algorithm.

**Figure 4 sensors-23-08295-f004:**
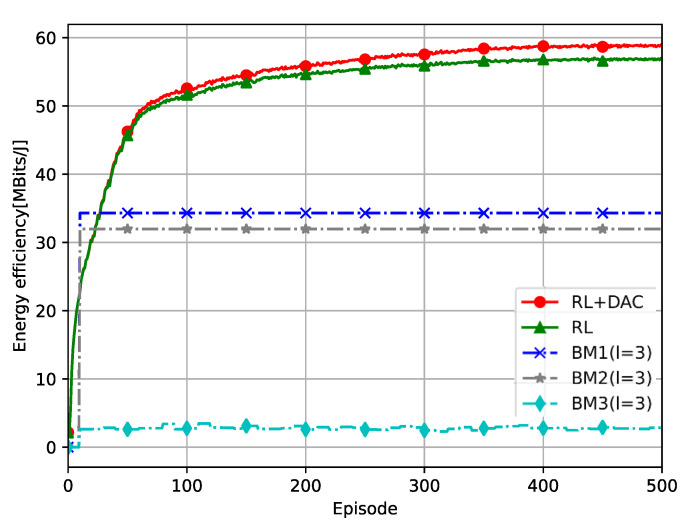
The convergence of Algorithm 1.

**Figure 5 sensors-23-08295-f005:**
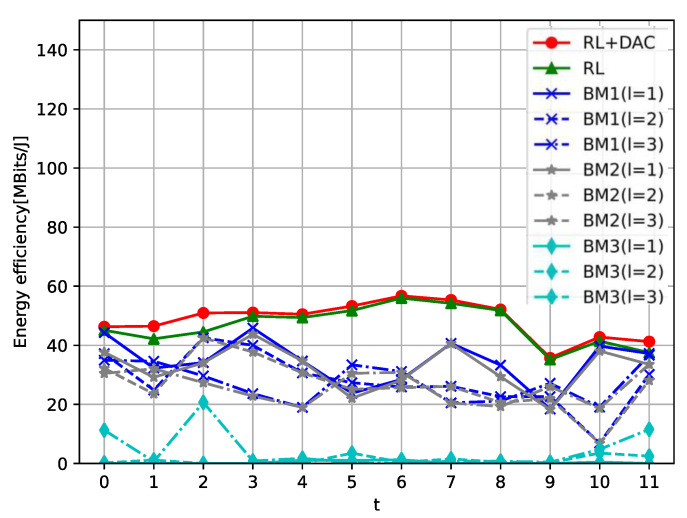
The total EE versus times.

**Figure 6 sensors-23-08295-f006:**
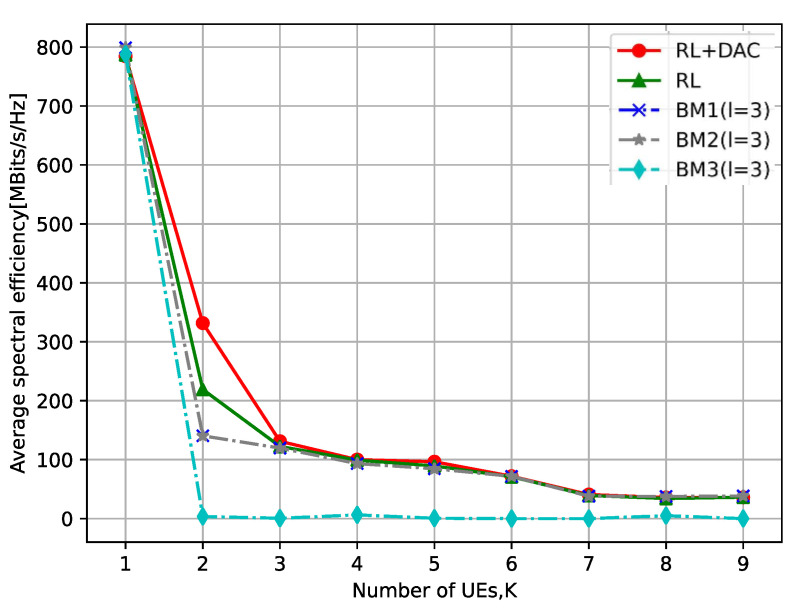
The relationship between the number of UEs and the average SE.

**Figure 7 sensors-23-08295-f007:**
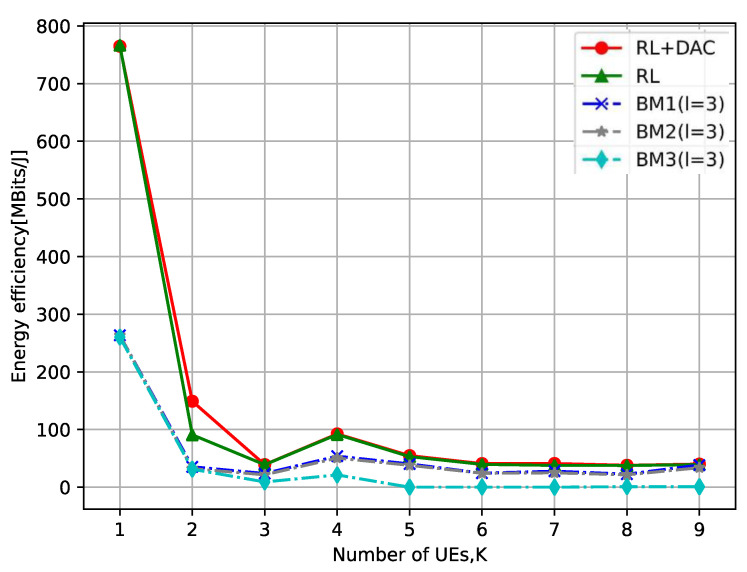
The relationship between the number of UEs and the total EE.

**Figure 8 sensors-23-08295-f008:**
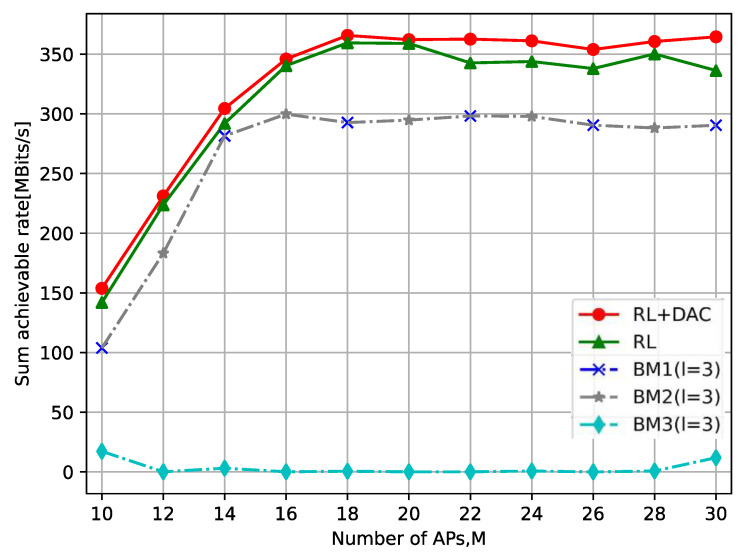
The relationship between the number of APs and the sum achievable rate.

**Figure 9 sensors-23-08295-f009:**
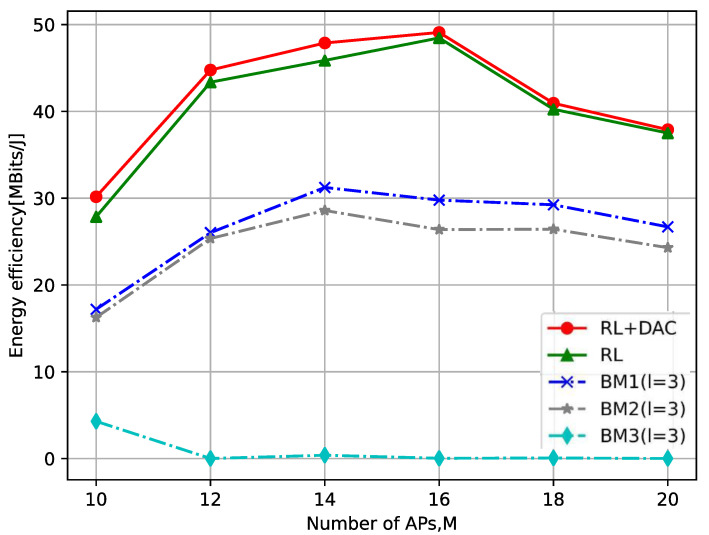
The relationship between the number of APs and the total EE.

**Figure 10 sensors-23-08295-f010:**
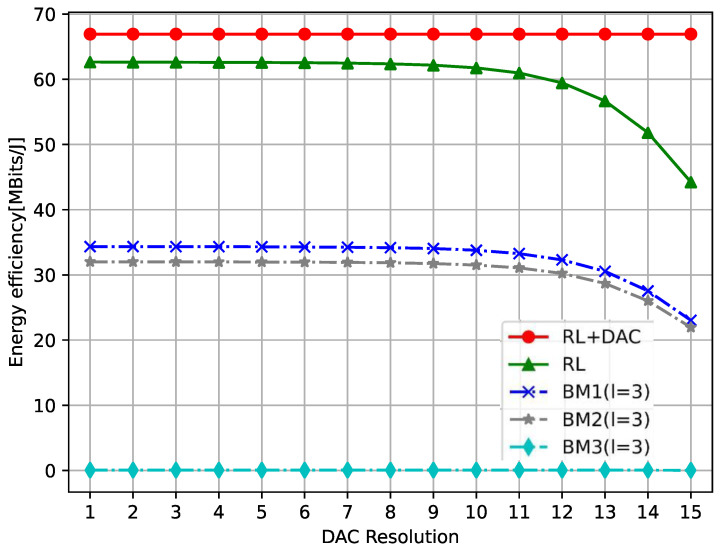
The total EE versus the DAC resolutions.

**Table 1 sensors-23-08295-t001:** The simulation parameters.

Parameters	Value
Bandwidth *B*	20 MHz
$\mathrm{Maximum}\;\mathrm{DL}\;\mathrm{transmit}\;\mathrm{power}\;P_{m}$	1000 mW
$\mathrm{Energy}\;\mathrm{consumption}\;\mathrm{of}\;\mathrm{fronthaul}\;\mathrm{link}\;E_{bh}$	$0.25\times 10^{-3}$ Joule/Mbit
$\mathrm{Energy}\;\mathrm{consumption}\;\mathrm{of}\;\mathrm{backhaul}\;\mathrm{link}\;E_{bb}$	$15E_{bh}$
$\mathrm{Thermal}\;\mathrm{noise}\;\mathrm{power}\;\mathrm{per}\;\mathrm{UE}\;\sigma_{w}^{2}$	$7.457\times 10^{-13}$ W
Path-loss exponent *α*	2
Zipf distribution factor *β*	1

## Data Availability

No new data were created or analyzed in this study. Data sharing is not applicable to this article.

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
