# Peer review of "Cluster Content Caching: A Deep Reinforcement Learning Approach to Improve Energy Efficiency in Cell-Free Massive Multiple-Input Multiple-Output Networks"

_sensors, 2023, doi:10.3390/s23198295_

Round 1
Reviewer 1 Report
This paper investigates a total energy efficiency model of cache-assisted CF-mMIMO system. An energy-efficient joint design of content caching, access point clustering and low-resolution digital-to-analog converter is proposed based on deep reinforcement learning. By utilizing the proposed DDPG algorithm, the simulation results verifies that the proposed scheme has significantly better performance than the traditional benchmark methods. There are some comments to improve the contribution of this work.
1. In the signal model, it may be better to present the link from the AP to the UE more clearly in Figure 1.
2. CSI acquisition is critical. This paper assumes perfect global CSI A discussion should be provided on it, e.g., the overhead to acquire the CSI, the impact of CSI error.
3. More discussions should be provided on the proposed DDPG algorithm. Why to choose this framework. What are the difference and advantage compared with the present works.
4. In the simulation, it is suggested to explain more clearly between RL+DAC and RL method. It is better to discuss this difference in detail.
5. As shown in Figure 4, the performance of the proposed scheme varies greatly along with t. It is suggested that give some explanation on Figure 4.
6. There exists repeated content in the discussion on Fig.8, on Page 13.
7. There are some typos, e.g.,
-The fifth line in the abstract, ‘ap’ should be upper-case.
-There are two Fig.3 in the paper.
-The line 209 in ‘3.2. Power Consumption’, ‘AP’ should be ‘APs’.
-The line 456 in the conclusion, the title should be ‘6. Conclusion’.
There are some typos, e.g.,
-The fifth line in the abstract, ‘ap’ should be upper-case.
-There are two Fig.3 in the paper.
-The line 209 in ‘3.2. Power Consumption’, ‘AP’ should be ‘APs’.
-The line 456 in the conclusion, the title should be ‘6. Conclusion’.
Author Response
The reply to the review report (Reviewer 1) is attached, please check it

Reviewer 2 Report
A cell-free massive multiple-input multiple-output system which cache has been investigated in the manuscript, where an energy-efficient joint design of content caching, access point clustering and low-resolution digital-to-analog converter has been presented based on deep reinforcement learning. The topic is interesting. My comments are as follows:
More recent work on related area should be mentioned in the introduction.
The complexity of the proposed algorithm should be further discussed and compared with benchmarks.
As the number of UEs increase, the average SE between RL+DAC and RL is almost the same. Why?
In Fig. 7, it seems that the sum achieveable rate for RL+DAC decrease when the number of APs is larger than 18. Why?
The manuscript could be further polished.
Author Response
The reply to the review report (Reviewer 2) is attached, please check it

Round 2
Reviewer 2 Report
I still concern about the sum achievable rate decrease for RL+DAC scheme in Fig. 8 (Original Fig. 7). As the number of AP increase, the diversity of the system is much higher. Is it possible to maintain the sum achievable rate when the number of AP is extremely large?
Minor editing of English language required
Author Response
The reply to the review report is attached, please check it.
